# Global within-species phylogenetics of sewage microbes suggest that local adaptation shapes geographical bacterial clustering

Marie Louise Jespersen [1,2], Patrick Munk [1], Joachim Johansen [2,3], Rolf Sommer Kaas [1], Henry Webel[2], Håkan Vigre [1], Henrik Bjørn Nielsen [3], Simon Rasmussen [2✉] & Frank M. Aarestrup [1✉]

Most investigations of geographical within-species differences are limited to focusing on a single species. Here, we investigate global differences for multiple bacterial species using a dataset of 757 metagenomics sewage samples from 101 countries worldwide. The within-species variations were determined by performing genome reconstructions, and the analyses were expanded by gene focused approaches. Applying these methods, we recovered 3353 near complete (NC) metagenome assembled genomes (MAGs) encompassing 1439 different MAG species and found that within-species genomic variation was in 36% of the investigated species (12/33) coherent with regional separation. Additionally, we found that variation of organelle genes correlated less with geography compared to metabolic and membrane genes, suggesting that the global differences of these species are caused by regional environmental selection rather than dissemination limitations. From the combination of the large and globally distributed dataset and in-depth analysis, we present a wide investigation of global within-species phylogeny of sewage bacteria. The global differences found here emphasize the need for worldwide data sets when making global conclusions.

[1] National Food Institute, Technical University of Denmark, Kgs. Lyngby, Denmark. [2] Novo Nordisk Foundation Center for Protein Research, Faculty of Health and Medical Sciences, University of Copenhagen, Copenhagen, Denmark. [3] Clinical-Microbiomics A/S, Copenhagen, Denmark. ✉email: simon.rasmussen@cpr.ku.dk; fmaa@food.dtu.dk

Sewage samples have proven useful for surveillance of antimicrobial resistance (AMR)[1,2] and infectious diseases, e.g. poliovirus, norovirus, and rotavirus[3,4]. Very recently, sewage samples have been used in the surveillance of the Covid-19 pandemic[5–7]. In supplement to such surveillance activities, understanding of the microbial community residing in sewage is important, because sewage has been suggested to comprise a reservoir of AMR and at the same time, provide an environment for potential genetic transfer between the bacteria in the community[8]. Several studies have examined the bacterial composition of sewage samples by 16 S rRNA investigations[9–11] or mapping to reference databases[12]. However, such investigations are limited to species previously identified and furthermore, it can be difficult to distinguish closely related species from 16S rRNA analysis, thus, species living in sewage that are closely related to species from the human gut could be confused. Within the last decade, investigations of microbiomes in human hosts, soil, plants, and more have found that differences in bacterial communities correlates with geography[11,13–16]. Among bacterial species isolated from clinical infections such as *Staphylococcus aureus*, *Streptococcus pneumoniae*, and *Escherichia coli*, within-species diversity correlating with geography has been observed in multiple studies[17–19]. Some of these geographical differences could be a result of local environmental selection but may also be due to the effect of dispersal limitations on local prevalence. These findings challenge the long-standing, Baas Becking ecological hypothesis that "Everything is everywhere, but the environment selects"[20].

In the attempt to disentangle the effect of environmental selection and/or dispersal limitations, researchers have studied the within-species diversity not only from isolates, but also in metagenomics data. Correlations between diversity within species and geography have been identified in bacterial species, such as *Eubacterium rectale* and *Candidatus* Pelagibacter, from marine metagenomes[21] and human gut microbiomes[22]. Another study did not identify significant geographical differences within subspecies of for instance *Bacteroides vulgatus* and *Alistipes putredinis* from the human gut[23]. However, the subjects included in that study were limited to North Americans and Europeans. It has often been difficult to obtain comparable samples in a standardized way across large geographies. The Global Sewage dataset[2], containing samples from 101 different countries, serves as an ideal candidate for a broader investigation of regional within-species diversity. A phylogenetic analysis of 79 samples focusing only on reference mapping to known bacterial species has previously been performed[12] and found geographical clustering for environmental and human commensal bacteria.

In this study, we aim to investigate the microbial community of sewage by constructing metagenome assembled genomes (MAGs) and determine the within-species phylogeny on a global scale, using 757 sewage samples from 241 sites and a total of 101 different countries. With these phylogenies, we increase the depth of the analysis by comparing geographical clustering between different genes when stratified by the cellular localization of the encoded proteins. Both in terms of sample size and global reach this study is the most comprehensive investigation of within species diversity among sewage species to date. From our analysis, we identified 3353 near complete (NC) MAGs from 1439 different MAG species and found that variation within some species (12 species out of 33 investigated) correlated with geographical separation. Furthermore, we found that, for a selection of species, genes associated with organelles displayed on average 10% less geographical variation compared to other groups of genes, suggesting that the geographical clustering is primarily due to environmental selection. Thus, we confirm the fundamental microbial ecology doctrine that microbes are globally dispersed but selected by the environment.

## Results

**Predominant bacteria in sewage do likely not originate from the human gut.** To identify bacterial genomes from sewage across the world, we used a combination of two different metagenomics genome binners (VAMB[24] and MetaBAT2[25]). From 757 samples across 101 different countries (Fig. 1a and Supplementary Fig. 1), we were able to create 3353 near complete (NC) metagenome assembled genomes (MAGs) assigned to a total of 1439 different MAG species. Of the MAGs we detected, 3301 were annotated to bacteria and 52 to archaea. The taxonomic distribution of the identified MAGs comprised 37 phyla, 75 classes, 151 orders, 259 families, 419 genera, and 215 species. However, we could only annotate 699 MAGs (20.8%) at species level, leaving 2654 unknowns. Likewise, there were unannotated MAGs at genus (29%), family (6%), and order (2%) level (see Supplementary Data 1 for complete taxonomic annotations). All MAG species were included in the analysis regardless of annotation level. The identified MAG species captured a wide range of taxonomy from the known microbial tree of life (Fig. 1b, c).

As the sewage was collected from urban areas, we were interested in knowing how large a fraction of the MAGs that could be associated with the human gut microbiome. The 3353 NC MAGs, we identified, were less than the number of NC MAGS (5036) found from binning of 1000 human faecal samples[24], however, these gut MAGs represented a lower number of different MAG species (645). Lower assembly quality and higher strain diversity has been suggested to reduce binning performance[26], which could be why the binning of the sewage metagenomes is more complicated than binning of the human gut samples. Additionally, we found that only 1.2% of all the NC MAGs could be identified as human gut microbiome species (mash distance <0.05 to any genome from the Unified Human Gastrointestinal Genome (UHGG) catalogue[27]), similar to a previous study based on mapping of reads from a subset of the Global Sewage samples, where 3.7% of the reads were found to be associated with the human microbiome[2]. In contrast to this, other studies using 16 S rRNA marker genes have found a higher proportion (15% and 4.3–28.7%)[10,11]. The difference in these results could be due to the limitations of each of the different methods for bacterial identification. There are advantages and disadvantages for different methods for bacterial identification and genome binning can be used to detect prevalent, including novel, bacterial genomes.

To further investigate differences in the bacterial composition, we compared the overall ratio of phyla between the seven World Health Organization (WHO) regions and a pool of human gut samples (Fig. 2a). The phylum-level composition based on the retrieved NC MAGs was similar in all samples regardless of geography with proportions of Firmicutes and Proteobacteria in all regions, consistent with untreated sewage 16S rRNA amplicon sequencing in Hong Kong and USA[9,11] and shotgun sequencing in Portugal[28]. However, the phylum compositions differed significantly from 1000 diverse human gut metagenomes (Two-sample Kolmogorov-Smirnov, $P < 2.2e{-}16$, Supplementary Data 2), suggesting again that the MAGs recovered from sewage do not primarily originate from the human gut. As the human gastrointestinal tract is a hypoxic environment, and sewage gets oxygenated, we expected an enrichment for aerophilic and aerotolerant organisms in the sewage system, which could drive a taxonomic shift. MAGs abundant in sewage indeed showed a higher prevalence of genes associated with oxidative phosphorylation, compared to MAGs abundant in humans (Fig. 2b).

When mapping all our reads back to the collection of classified MAGs from this study and human gut, we confirmed similar regional taxonomic profiles (Fig. 2c). Additionally, these results suggest that species associated with humans make up a minor

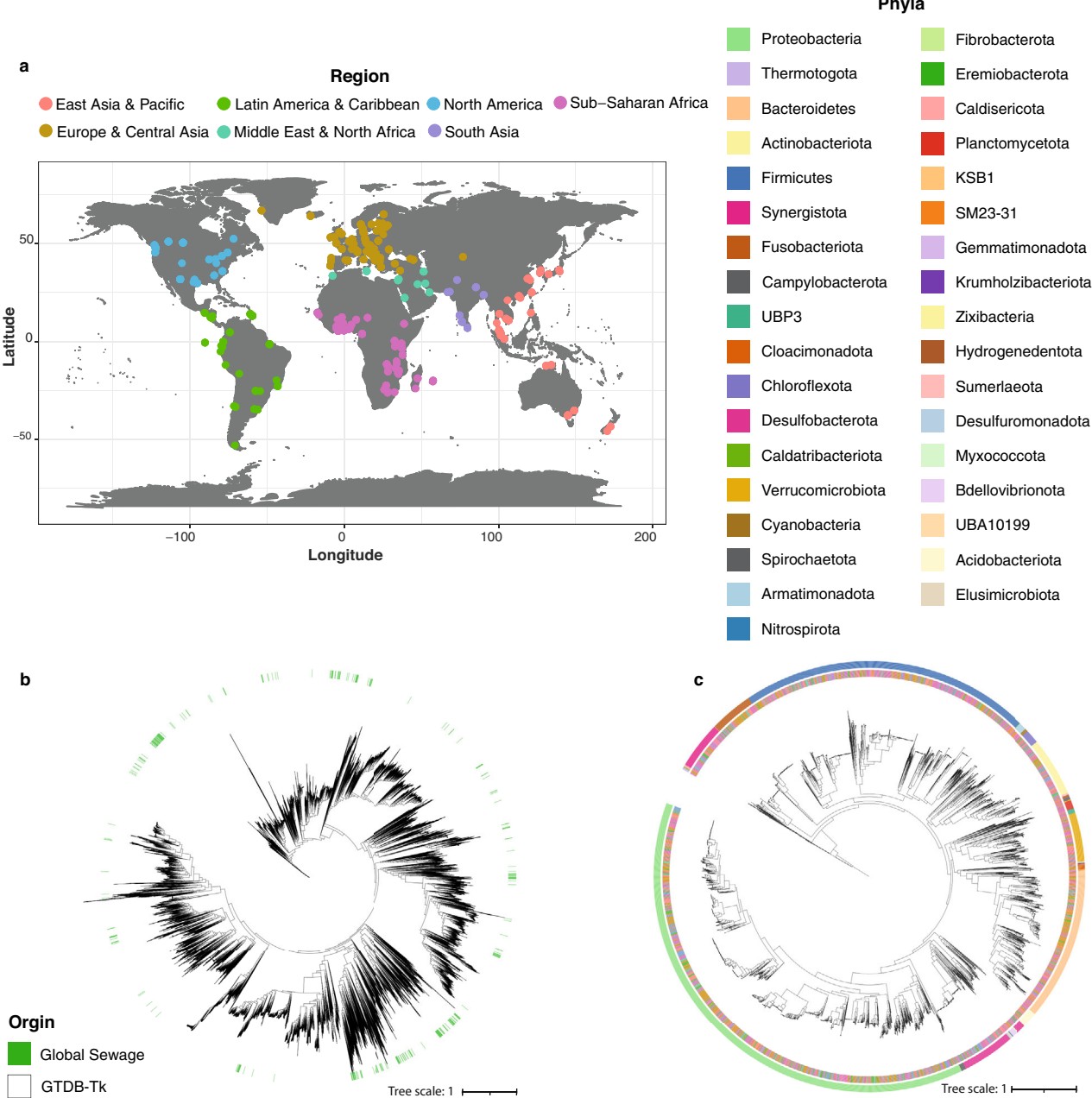

**Fig. 1 Distribution of samples and MAGS worldwide. a** World map of sampling sites. 757 sewage samples were collected from 241 different sampling sites spanning 101 different countries. Sampling sites are highlighted and coloured according to the regional grouping from the World Health Organization (WHO). Sampling times can be found in Supplementary Fig. 1. The map was created using ggplot2 in R[61]. **b** Maximum likelihood (ML) tree of marker gene, amino acid alignments for all bacterial MAGs and bacterial genomes included in GTDB-Tk. The MAGs identified from sewage are scattered throughout the tree of known bacterial species. **c** ML tree of marker gene, amino acid alignments for all bacterial MAGs identified in this study. The identified MAGs are clustered according to phyla rather than geographical origin. Inner band is showing the geographical origin of samples according to WHO region, colours follow legend in b. Outer band is showing the phyla of MAGs, coloured according to the legend on top.

proportion of a taxonomically more diverse sewage bacteriome with more potential sources (mean Shannon diversity of >3.75 and 2.4 in sewage and human gut, respectively, Fig. 2d). Furthermore, around 70% of the human gut microbiome reads, whereas just 41% of the sewage reads were assigned to the MAG collection.

Together our results suggest a major shift in taxonomy, enrichment for aerophilic organisms and a lower genome retrieval efficiency, due to higher alpha diversity and multi-host strain heterogeneity, in the sewage compared to human guts.

**Sewage bacteria vary according to geography**. To infer the phylogeny and identify geographical clustering within single species, we identified MAG species that were present across multiple samples. From the 1439 different MAG species, only 41 contained MAGs retrieved from ten or more different samples. Altering the 95% species Average Nucleotide Identity (ANI) threshold to 90% or 97.5% only resulted in a few variations to the originally identified MAG species cluster sizes and only a single new MAG species, found in ten or more samples, was identified (Supplementary Fig. 2). This is consistent with other studies

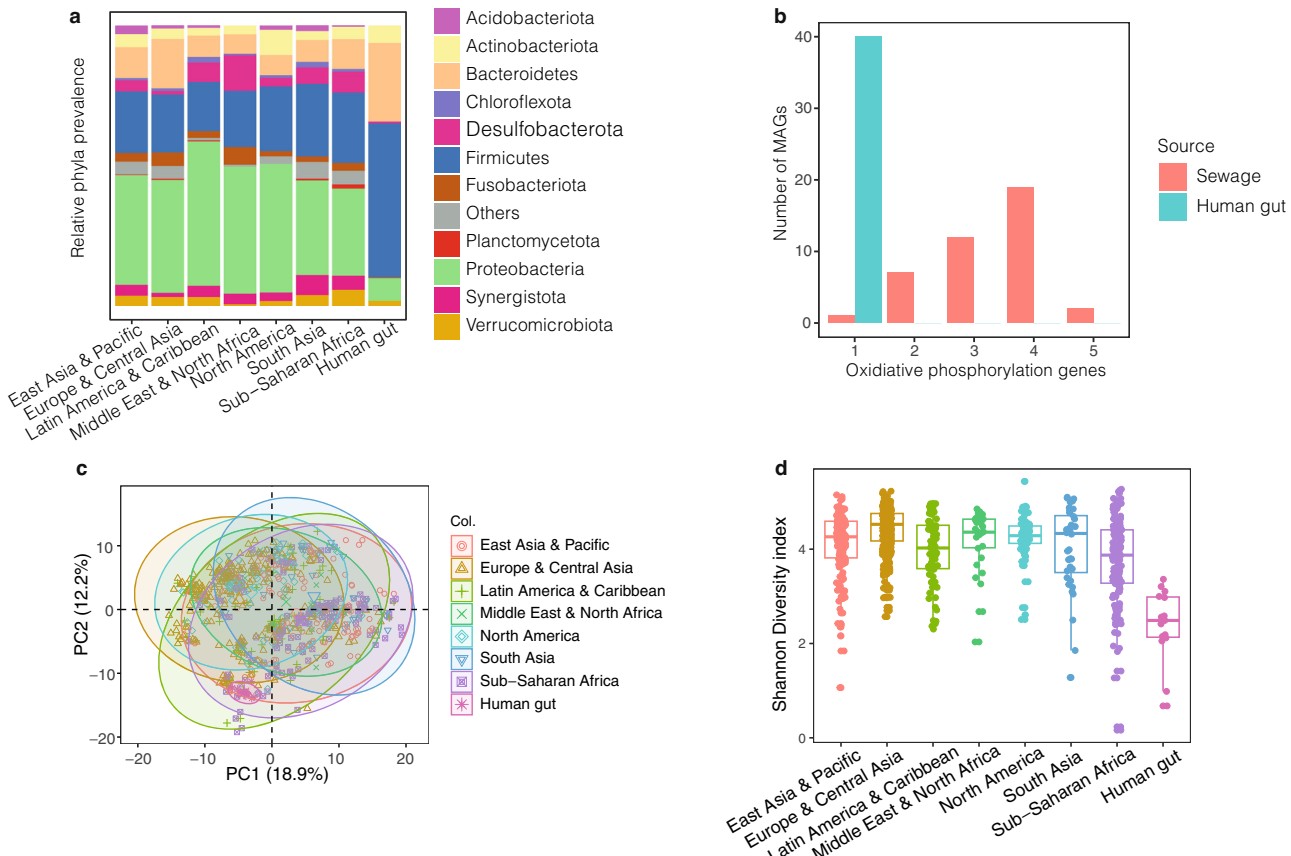

**Fig. 2 Bacterial composition of sewage. a** Relative frequency of the 11 most prevalent phyla between the different regions. The ratio plotted is the taxonomy of the combined pool of MAGs from all samples of a certain region. In this plot, only the 11 phyla found in more than ten samples across regions were plotted, while the remaining MAGs were grouped into one combined category (others). The phyla ratio between sewage samples of all regions were similar, but the phyla distribution in the human gut samples were different from these (Two-sample Kolmogorov-Smirnov, $P < 2.2e−16$, Supplementary Data 2). **b** Frequency of genes involved in the oxidative phosphorylation pathway. These genes were identified from the best representative genome of the 40 most prevalent MAG species from the Global Sewage data set and from the MAG species obtained in Nissen et al. 2021[24]. Colours according to sample origin of the MAG species genome. Only one gene involved in oxidative phosphorylation is found in all human gut MAG species, whereas up to five genes from this pathway is found in the MAG species from sewage. **c** PCA plot of clr transformed read count abundance of the combined data set of 2,084 MAG species obtained in this work and in Nissen et al. 2021[24]. The bacterial abundances (phylum level) in sewage samples across all regions were similar, however, the bacteria in the human gut samples consists of only a small fraction of the variety of bacteria found in sewage. **d** Shannon Diversity index comparison between sewage samples from different regions and human gut samples. The Shannon diversity index was calculated from Transcripts Per Kilobase Million (TPM) using the vegan package in R[58]. The alpha diversity is similar between sewage samples from all regions, but the diversity in the human gut samples is lower. Sample sizes for figures (**a**, **c**, and **d**) are $n = 244, 136, 71, 160, 80, 36, 30$, and 15 for Europe & Central Asia, East Asia & Pacific, Latin America & Caribbean, Sub-Saharan Africa, North America, South Asia, Middle East & North Africa, and Human gut respectively. The boxplot centre, lower and upper hinge correspond to the median, first and third quantiles, respectively. The upper and lower whiskers extend to the largest and smallest values, no more than 1.5* the inter-quartile range (IQR, ie. the distance between first and third quartiles). Data points beyond these values are plotted as individual outliers.

suggesting the use of 95% ANI as a species threshold[29,30], however, ANI boundaries could vary between species and these results are debated[31,32].

For each of the 41 species of 10 or more genomes, we identified orthologous genes and found between 1437 and 4967 different orthologous genes for each species. We inferred a maximum likelihood tree for each gene and created a combined multi-gene phylogenetic tree for each species using ASTRAL. For instance, Cluster 5 (C5), which we identified as belonging to the *Brachymonas* genus, contained between 1 and 12 MAGs from each geographical region and the phylogenetic species tree was based on 2298 orthologous gene trees (Fig. 3a). Potential chimeric genomes could be placed incorrectly in the phylogenetic trees, to validate that our approach was not biased by such chimeric genome bins, we repeated the species-level phylogenetic trees with a dataset of 50 human gut samples spiked with 1–3

different *Salmonella* strains from Nissen et al. 2021[24]. We found that only very similar strains (ANI > 99%) were in risk of being mixed into one MAG and that such mixed MAGs were placed exactly as or very similar to the most identical reference strain in the species trees, thus not affecting the results from our analyses (Supplementary Fig. 3 and Supplementary Table 1). Additionally, we checked if the within-species differences found in the trees were biased by the binning methods (VAMB vs MetaBAT) and found that this was not the case ($R^2 < 0.25$, Supplementary Fig. 4).

To assess the degree of geographical clustering and potential dispersal limitation, we performed a PERMANOVA test on the distance matrices from the ASTRAL trees with more than one genome from at least two regions, leading to a total of 33 tested species (Supplementary Fig. 5). Geographical clustering was identified as the $R^2$ value from the PERMANOVA test, which

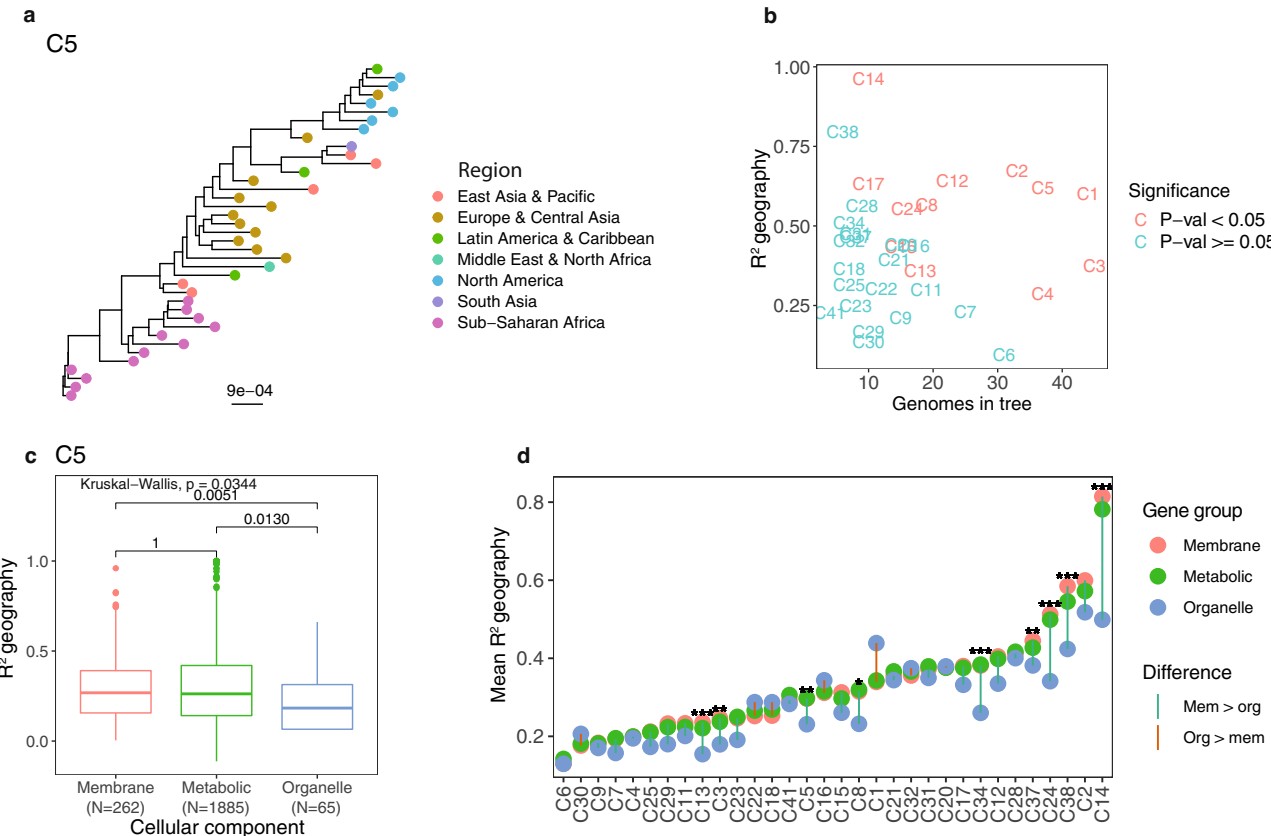

**Fig. 3 Geographical clustering within species. a** ASTRAL species tree of *Brachymonas*, Cluster 5 (C5). This tree was created from 2,298 orthologous gene trees. The tips in the tree are coloured according to WHO regions. PERMANOVA testing of this tree gave a geographical $R^2$ of 0.62 and a $p$ value < 0.001. **b** Geographical clustering of 33 MAG species. Results from PERMANOVA testing on ASTRAL species trees; the geographical $R^2$ values are plotted against the number of MAGs in the tested tree. MAGs that were the only representative of a region in a tree were excluded from the tree prior to PERMANOVA testing. Additionally, trees could only be tested if MAGs from at least two different regions were present in the tree. Points were coloured according to significance level of the PERMANOVA $p$ value after Benjamini-Hochberg (BH) correction. In 12 ASTRAL species trees the MAGs were significantly clustered according to regional origin (Supplementary Data 2). **c** Geographical clustering in different cellular component groups of *Brachymonas* (C5). The geographical $R^2$ values of *Brachymonas* (C5) at the y axis divided by the different gene groups (membrane, metabolic, and organelle) at the x axis. We tested for significant differences between the groups with a Kruskal-Wallis Rank Sum Test ($P = 0.0344$, BH corrected), and afterwards for significance levels between gene groups with a Pairwise Wilcoxon Rank Sum test ($P = 1$, 0.0051, and 0.0130, BH corrected). $P$ values from both tests were BH corrected. The boxplot centre, lower and upper hinge correspond to the median, first and third quantiles respectively. The upper and lower whiskers extend to the largest and smallest values, no more than 1.5* the inter-quartile range (IQR, ie. the distance between first and third quartiles). Data points beyond these values are plotted as individual outliers. **d** Geographical clustering in different cellular component groups of all tested MAG species. Mean geographical $R^2$ values from PERMANOVA testing on gene trees in each cellular component group plotted for all 32 tested MAG species. Differences in geographical $R^2$ values between groups, where tested like described previously with a Kruskal-Wallis Rank Sum Test and subsequent Pairwise Wilcoxon Rank Sum test. The significance level highlighted is of the $p$ value between genes from organelles and membrane from the Wilcoxon test adjusted by BH correction. We found the geographical clustering ($R^2$) of the organelle genes to be lower than the ones from the membrane genes in 25 MAG species, of which nine were significantly lower. The distribution of gene tree geographical $R^2$ values and sample sizes can be found in Supplementary Fig. 9. *$P < 0.05$, **$P < 0.01$, ***$P < 0.001$.

describes how much of the variation in the data that can be explained from the regional origin of the samples. Of the 33 tested trees, 12 clustered significantly ($P < 0.05$, BH corrected) according to genome sample region (Fig. 3b). These were the trees of *Lactococcus raffinolactis*, *Brachymonas denitrificans*, *Escherichia coli*, and *Escherichia flexneri* and unknown species within the genera: *Neisseria*, *Veillonella*, and *Brachymonas*, the families: *Leptotrichiaceae*, *Desulfobulbaceae*, and *Saccharofermentanaceae*, and the *Selenomonadales* order. These results were consistent when pruning trees from outliers (Supplementary Fig. 6). Other metagenomic studies of sewage and human gut samples have also shown that some species vary according to geographical dispersion, while others do not[12,22,24]. One study of the human gut microbiome found a higher number of species within

*Firmicutes*, where variation was geographically separated[22]. However, we did not find any phyla enriched in geographical clustering (Supplementary Fig. 7). On average 56% of the variation in the 12 significant trees could be explained by geography, additionally, several smaller clusters (C26, C28, C34, and C38) with a high degree of geographical clustering (>50%) did not achieve significance (0.11 > $p$ > 0.05). The portion of variation explained by geography is much higher than the average of 19% variation in the significant trees found by a similar study of the human gut microbiome[24], suggesting that more of the genomic variation found in environmental bacteria correlates with geography than the variation found in gut microbes. The geographical clustering suggests that either dispersal limitations or local selection can cause localized divergence.

**Organelle-associated genes show less geographical variation**.
To investigate whether the observed geographical clustering was
the result of dispersal limitations or local selection, we examined
whether the clustering varied between different groups of genes.
We hypothesised that proteins on the surface of the bacteria
would be subject to selective pressure from the environment,
because of their direct interaction with the surroundings,
whereas the intracellular proteins, not directly interacting with
the environment, would vary as a consequence of genetic drift.
We therefore tested whether the variation explained by geo-
graphy differed as a function of overall gene product cellular
location (organelles, membrane, and metabolic). As we found
that organelle genes varied less than other genes in 94% of
species (Supplementary Fig. 8), we wanted to ensure that our
test was not driven by differences in gene conservation levels.
We found that this was not the case, as gene variation was
unrelated to geographical clustering ($R^2$) (Supplementary
Table 2). Additionally, gene lengths, regional distributions
across trees, and the number of MAGs per species were also
unrelated to geographical clustering ($R^2$), with the exception of
one species (C26), which we therefore excluded from further
analysis.

With a Kruskal-Wallis Rank-Sum test on geographical
clustering ($R^2$) from the three gene groups (membrane,
organelle, and metabolic), we found nine MAG species with
significantly ($P < 0.05$, BH corrected) different geographical
clustering ($R^2$) between the groups after Benjamini Hochberg
(BH) correction of $p$ values. The significant MAG species
included *E. coli*, *E. flexneri*, *Sebaldella termitidis* and unknown
species in the *Veillonella*, *Brachymonas*, and *Neisseria* genera and
the *Desulfobulbaceae*, *Saccharofermentanaceae*, and *Thiotricha-
ceae* families. One of the MAG species with significant
differences was C5 (*Brachymonas*), in which the trees of genes
associated with organelles had significantly lower geographical
clustering ($R^2$), than trees of genes from the other two groups
(Fig. 3c). The lower geographical clustering ($R^2$) in organelle
gene trees were also found for the eight other significant MAG
species (Fig. 3d and Supplementary Fig. 9) (two-sided Wilcoxon
Rank Sum test, $P < 0.05$ BH corrected, Supplementary Data 2).
Moreover, six of the nine significant MAG species were
significantly clustered according to geography in the ASTRAL
species tree and the three remaining clusters (C34, C37, and C38)
showed a high degree of geographical clustering but were not
significant. Likewise, we saw lower geographical clustering ($R^2$)
in organelle genes for the majority (16 of 23) of the remaining
MAG species without significant difference.

To support these findings, we calculated the dN/dS ratio for the
nine MAG species with significant differences between organelle
and membrane geographical clustering ($R^2$). For all the nine
MAG species, the dN/dS ratios were larger for membrane genes
than organelle genes, suggesting positive selection of the
membrane genes (Supplementary Fig. 10). Furthermore, this
difference was significant for seven MAG species (C3, C5, C13,
C14, C34, C37, and C38) (two-sided Wilcoxon Rank Sum test,
$P < 0.05$, BH corrected, Supplementary Data 2).

Collectively, our results show similar levels of geographical
clustering in the membrane genes and metabolic genes for a
selection of species, whereas the genes associated with the
organelles displayed significantly less geographical clustering. We
expect that the organelle genes would have followed a similar
evolutionary trajectory as the metabolic and surface genes if the
clustering was a result of dispersal limitations. Our results thus
suggest that for some species (36% of the investigated) the
geographical clustering is primarily due to regional selection,
rather than dispersal limitations.

## Discussion

Here, we present an investigation of how sewage bacteria are
dispersed globally. We found that in general the bacteria identi-
fied by metagenomic binning in untreated sewage, from the inlet
to the wastewater treatment plants, showed a degree of geo-
graphical clustering for the within-species diversity. Interestingly,
the genomic variation in organelle genes showed less regional
clustering than genes involved in metabolic and membrane
functions, suggesting that the clustering observed is primarily due
to selection rather than dispersal limitations. Thus, the bacteria
residing in sewage can spread globally, but are under evolutionary
pressure to adjust to the different environments across the world.
This selection pressure combined with the co-existence of mul-
tiple bacterial species and the presence of antimicrobial resistance
genes (ARGs) and antimicrobial drugs in sewage create a high
probability for transferral of ARGs between bacteria[8]. To prevent
global transmission of these genes, it is important to better
understand how sewage bacteria are globally disseminated.

The WHO has a goal of delaying the dissemination and emer-
gence of AMR through monitoring[33]. We have previously suggested
that sewage sampling is a desirable strategy for such surveillance
activities[34]. Metagenomics binning can be used to identify novel,
bacterial genomes that are not present in reference databases. Here,
we found that binning of shotgun sequences could identify a frac-
tion of the bacteria residing in the sewage samples and that the
bacteria originating from the human gut were a small subgroup of
the microbiome found in these samples. This is a reminder that
when using sewage samples for surveillance activities, the detection
method should be selected carefully. If monitoring human-derived
pathogens in sewage, one needs to consider that DNA from these
organisms is a very small fraction of the total pool of microbial
DNA. Mapping to reference genomes can be useful for the mon-
itoring of the global spread or the local levels of a particular species,
like we have seen for the covid-19 pandemic[7]. However, metage-
nomic assembly and binning could be used to identify potential
candidates for surveillance and possibly to clarify the genomic
context in which ARGs emerged and disseminate. Thus, there is
more potential for new and important discoveries using metage-
nomics binning of sewage samples.

In conclusion, this study shows a clear geographical phyloge-
netic clustering of 12 bacterial species from sewage and suggests
that this could be caused by global differences in the selection
pressure in wastewater and the corresponding adaptation of
sewage bacteria. Thus, demonstrating the importance of more
understanding of the dynamics in the microbial life residing in
sewage. Furthermore, the worldwide differences found with-in
different bacterial species underpin the importance of including
samples from the entire world if global conclusions must be made.

## Methods

**Global sewage dataset**. Samples were collected and handled as part of the Global
Sewage project[2,35]. In brief, untreated sewage samples were collected before the inlet
to the wastewater treatment plant at sample sites. DNA was extracted and frag-
mented from the untreated sewage and libraries were sequenced using Illumina
paired-end sequencing to an average sequencing depth of approximately 42 mio
reads. BBduk was used for adaptor removal and quality trimming of reads, using a
quality threshold of 20 and a minimum read length of 50 bp. In total, 757 samples
from 241 sites spanning 101 different countries across the world were included in
this project. Of the 757 different samples, 56 were re-sequenced to reach a sufficient
sequencing depth, the handling of these duplicates is described under the relevant
methods. A complete list of samples included can be found in Supplementary Data 3.

**Genome binning**. We assembled forward, reverse, and singleton reads with
metaSpades (v3.13) using kmer sizes between 27 and 127 bp with an interval of
20 bp. Scaffolds above 1000 bp were saved for further analyses. For binning with
MetaBAT2 (v2.10.2)[25], we filtered contigs to a minimum size of 1500 bp and
performed single-sample binning with MetaBAT2 using default settings. For bin-
ning with VAMB (v3.0.1)[24], we combined contigs >2000 bp from all samples into

one catalogue. From a pilot run with VAMB[24] binning, we found no increase in the number of NC MAGs when using contigs >1500 bp, therefore we chose a minimum contig size of 2000 bp to reduce mapping time. We mapped reads from each sample to the contig catalogue using Minimap2 (v2.6)[36]. Afterwards, jgi_summarize_bam_contig_depths from MetaBAT (v2.10.2)[25] was used to calculate abundances of contigs in each sample. The output abundances were combined into a matrix, normalized using vambtools[24], and used as input to VAMB. We calculated and normalized Tetra Nucleotide Frequencies (TNFs) using vambtools as well. From the contig catalogue, we obtained contig names and lengths and used them as input to VAMB along with the normalized TNFs. We ran VAMB using the memory mapping mode available at the github repository. VAMB was run using a GPU with a mini-batch size of 256 and a network of 48 latent and 1024 hidden neurons.

We assessed the quality of all MAGs with a size above 1 Mbp using CheckM (v1.1.3) lineage_wf[37]. We defined the quality of MAGs as in Almeida et al. 2019[38] where NC MAGs were defined as >0.9 completeness and <0.05 contamination. We used NC MAGs in our further analysis. To group similar MAGs (likely to be different variations of the same bacterial species) into clusters, we used dRep compare (v2.2.3)[39] with a mash threshold of 90% and an Average Nucleotide Identity (ANI) threshold of 95% on all identified NC MAGs. dRep dereplication was run with the same thresholds on each of the dRep clusters identified from the comparing and the resulting score was used to select between MAGs from the same sample within one cluster, to avoid redundancy if a bin was identified both by MetaBAT and VAMB. Additionally, the dereplication results were used to select the best MAG species representative for each cluster of MAGs. We assessed the taxonomy of all NC MAGs with GTDB-Tk classify_wf (v0.3.2)[40].

For comparison of the taxonomic distribution of sewage MAGs to human gut species taxonomy, the MAGs created in Nissen et al. 2021[24] were used. Difference in phylum distributions between human and sewage samples were tested for significance with a two-sample Kolmogorov-Smirnov test in R. Additionally, these human gut MAGs were dereplicated like the MAGs from the sewage data and used together with reads from 15 randomly selected human samples (no infant or diseased hosts) from the dataset from Almeida et al. 2019[38] for abundance comparisons. The read count abundances of MAG species were obtained using CoverM (v 0.6.1)[41], by mapping to the best representative MAG species genomes. In the mapping process reads from duplicated samples were pooled into a single fastq file, and prior to abundance investigations the counts from all samples were rarefied. The rarefy function from the Vegan R package was used to rarefy the counts to the minimum number of counts (431.564) found in a sample, afterwards the abundance matrix was filtered based on the expected/covered ratio of a genome (>=0.5) and Transcripts Per Kilobase Million (TPM) were calculated from the read counts. The expected coverage (c) of a genome was calculated as:

$$c = 1 - \left(1 - \frac{l}{g}\right)^r \quad (1)$$

Where l is the average read length, g is the genome length, and r is the number of counts mapped. This calculation is simplified from the calculation described in Rasmussen et al. 2015[42]. The fraction of sewage MAGs likely to originate from the human gut microbiome was found by using MASH (v2.0)[43] to map to the Unified Human Gastrointestinal Genome (UHGG) catalogue[27] and identified as within a mash distance of 0.05 to any genomes in this catalogue.

**Phylogeny.** We reconstructed phylogenetic trees containing all the identified MAGs using the marker gene set from GTDB-Tk. One tree was created including the GTDB-Tk reference species and another without these. Both trees were inferred with FastTree (v2.1.11)[44] and rooted on a *Thermotogae*, because this is the bacterial phylum most closely related to Archaea[45]. We visualized the trees using iTol (v1.0)[46]. For the 41 dRep clusters spanning ten or more samples, a separate tree of the MAGs belonging to each cluster was inferred. For this, we used Prodigal (v2.6.3)[47] protein predictions from GTDB-Tk as input to Sonicparanoid (v1.3.4)[48], to identify orthologous genes, using the fast mode. To align DNA sequences of all identified orthologous, we used MAFFT (v7.453)[49]. Samples were excluded from the alignments if they had more than one copy of an orthologous gene, to avoid uncertainty of which gene was used to infer phylogeny. We used TrimAl (v1.4)[50] to convert alignments to phylip format, prior to building a separate phylogenetic tree for each gene using IQ-TREE (v1.6.8)[51] with automatic model selection. Trees were created if a gene was observed in at least three samples, which is the lowest possible number of samples that a tree can be inferred from with IQ-TREE. To infer the overall species tree phylogeny, we used all the gene trees from a specific MAG species as input to ASTRAL (v5.7.4)[52]. In this tree, IQ-TREE was used to correct branch lengths with the ASTRAL tree as constrained tree input. We used the ggtree (v2.0.4) package in R[53] for visualization of species trees. To validate this method for investigation of within-species differences, we applied CheckM and dRep compare with same settings as described to a dataset of 50 human gut samples spiked with one to three different *Salmonella* strains and furthermore, created an ASTRAL species tree as described above for the MAG species found from this analysis.

**Functional annotation.** To assign functional annotation to the genes, we used the Prodigal protein predictions from GTDB-Tk as input to InterProScan (v5.36-75.0)[54]. From the InterProScan output, we then extracted the GO-term annotation and used the GO.db-package (v2.1)[55] in R to get the annotations within the Cellular Component category. We grouped the genes into the top-level annotations within this category, to get overall groupings for comparisons between gene groups. We selected the groups membrane and organelle for further analysis, and the remaining genes were combined into one collapsed group. The organelle-associated genes were mostly coding for proteins that were a part of the ribosome (85%) and to a lesser extend proteins bound to the chromosome (9%), acting as part of the flagellum (5%), or polyhedral organelles (1%). For many of the orthologous genes (up to 91%), it was not possible to annotate them to any cellular component GO term. To include these unannotated genes in the analysis, we grouped them with gene groups other than membrane and organelles. When investigating the biological process GO term annotation of this gene group, the largest fraction (on average 45%) of genes were annotated to be part of a metabolic pathway and we therefore considered this group to represent metabolic genes. The genes involved in oxygen tolerance were likewise identified from the InterProScan output, by identifying the genes involved in the KEGG pathway map00190, oxidative phosphorylation.

**dN/dS calculation.** Codeml (paml (v.4.9j)[56]) was used for genewise dN/dS calculation. Genes with genetic variation between samples were identified with the snppos_analyzer from CSI phylogeny (v1.4)[57] and only these genes were input to codeml. Furthermore, to make sure that gene alignments were in frame, only alignments starting with a start codon (ATG, TTG, or GTG) were included. The phylip format gene alignments from the phylogeny reconstruction were converted to fasta files using TrimAl (v1.4)[50] and stop codons were removed prior to dN/dS calculation. Along with the alignments, the gene tree files from IQ-TREE (v1.6.8)[51] were used as input to codeml paml (v.4.9j)[56]. One dN/dS ratio per gene was calculated with codeml by setting the model option to 0 and the seqtype option to codons. Additionally, optimization was performed one branch at the time (method: 1) and ambiguous sites were removed from the calculation (cleandata: 1), otherwise default settings were used.

**Statistics and reproducibility.** To determine the amount of geographical variation for both GTDB-TK tree, species trees, and gene trees, we used the adonis2 function from the vegan package in R (v2.5–6)[58] to perform a Permutational multivariate analysis of variance (PERMANOVA) according to geography. Geographical clustering was identified as the $R^2$ value from the PERMANOVA test, which describes how much of the variation in the data that can be explained from the regional origin of the samples. Prior to the testing, multiple MAGs from the same city within one tree were limited to one representative MAG based on the dRep score, this filtering also excluded any duplicate MAGs from duplicated samples. In addition to this, a MAG was removed from the tree if it was the only representative of a region in this tree. To adjust for multiple testing, we corrected the p values from these tests using Benjamini & Hochberg (BH) algorithm[59], adjusting for the number of ASTRAL species trees tested with PERMANOVA. For some short genes with low variance, the geographical $R^2$-values outputted from the PERMANOVA test were negative (Supplementary Fig. 11). It is possible to get a negative $R^2$ value when the fitted model is worse than a horizontal line.

To identify species with any significant differences in geographical $R^2$- or dN/dS-values between gene groups, we grouped the values of the gene trees according to the Cellular Component annotations and used a Kruskal-Wallis test on the values from the different groups. Afterwards, we applied a Wilcoxon Rank Sum Test on the groups from MAG species displaying significance ($P < 0.05$, BH corrected) from the Kruskal-Wallis test, to identify which of the three groups that were differing from each other. P values from the Wilcoxon- and Kruskal-Wallis tests were adjusted using the BH algorithm[59], adjusting for the number of gene group tests.

To support the comparisons of geographical clustering ($R^2$) between gene groups, we investigated if different gene qualities could bias the geographical $R^2$ value. This was done by calculating the Pearson Correlation Coefficient (PCC) for geographical $R^2$ values according to the number of samples in the tree, gene variation, gene length, and regional entropy. Number of MAGs was counted in a tree after removing duplicate city and single region samples, as it was done prior to the PERMANOVA. Gene variation was obtained as the mean fraction of varying sites across all pairwise sequences (mean pi). Gene lengths were identified as the number of positions in the fasta output from Sonicparanoid. Regional entropy was calculated as:

$$-\sum_{i=1}^{n} ln(p_i^{p_i}) \quad (2)$$

where n was the total number of regions in the tree, and $p_i$ was the proportion of samples belonging to a specific region. We performed these calculations on all gene trees tested with PERMANOVA. P values, test statistics, and 95% confidence intervals of Kruskal-Wallis-, Wilcoxon-, and species tree PERMANOVA tests can be found in Supplementary Data 2.

**Reporting summary**. Further information on research design is available in the Nature Portfolio Reporting Summary linked to this article.

## Data availability

The raw reads are available in the European Nucleotide Archive (ENA) under the accession numbers: PRJEB40798, PRJEB40816, PRJEB40815, PRJEB27621, and ERP015409[2,35]. Source data for Fig. 2a, b and d are found in Supplementary Data 1, while source data for Fig. 3c, d are in Supplementary Data 2. Other data are available upon reasonable request.

## Code availability

The code used in this paper is available on GitHub at https://github.com/marieljespersen/Sewage_MAG_phylogeny [60].

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

## Acknowledgements

We thank Anders Gorm Pedersen for a fruitful discussion on the phylogenetic analysis. This work was supported by The Novo Nordisk Foundation (NNF16OC0021856: Global Surveillance of Antimicrobial Resistance). S.R., J.J., and H.W. was supported by the Novo Nordisk Foundation (grants NNF14CC0001, NNF20OC0062223 and NNF19SA0059348).

## Author contributions

F.M.A. and S.R. conceived the idea and guided the analyses. M.L.J performed the analysis. P.M. performed the metagenomics binning with MetaBAT2. P.M., J.J., H.V., H.W., R.S.K., H.B.N., S.R., and F.M.A. provided guidance for the analysis and input for interpretation of results. M.L.J., S.R., and F.M.A. drafted the paper with contributions from all co-authors. All authors have read and accepted the final version of the manuscript.

## Competing interests

H.B.N. and J.J. are employed at Clinical-Microbiomics A/S. The additional authors declare no competing interests.
