## [Peer Review File · Communications Biology]

This manuscript has been previously reviewed at another Nature Portfolio journal. This document only contains reviewer comments and rebuttal letters for versions considered at Communications Biology.

Reviewers' comments:

Reviewer #2 (Remarks to the Author):

I reviewed this article previously for Nature Micro. I think the article is better suited to Communications Biology and will be of interest to the readership. I am satisfied that the authors have addressed most of my comments from the previous round of review. Although I do think the text could be further streamlined to assist in readability. Further comments follow.

Given much is made of the significance of this work (in gaining an insight into dispersal mechanisms for sewage related microbes) in the study of the spread of ARGs, I am curious as to why the authors didn't look into this here at all. I think many readers will find this strange.

Line 83 - Define NC and MAG.

Line 87 - Which version of GTDB was applied? The later versions give species annotations. If these versions were used, is it the case that only 20% received annotation to this level?

Lines 103-110 - I found the logic here to be vague and hard to follow. It is also likely due to the resolution differences between 16S and metagenomics and the cut-off values for similarity selected. Difficult to compare a figure like this.

Lines 112-144 & 146-181 - Long paragraphs and in places hard to follow. Consider breaking up and streamlining the text.

Lines 199-200 - This sentence is not well integrated.

Genus names should start with a capital letter and be italicized. i.e. *Salmonella*.

Line 235 - I don't think "comprehensive" is the right word here. Although a lot of work was done in this study, to me comprehensive would have involved more than just inference from metagenomic analyses. I would remove/rephrase.

Lines 261-262 - "caused by selection pressure" is a bit vague. I think this should be rephrased.

Lines 581-582 - Is this not the other way around? i.e. Only one gene in the human...

Line 596 - I found the interchangeable use of C5 and *Brachymonas* here to be confusing. Also, in the main text you have spelled it "Brahymonas".

Reviewer #3 (Remarks to the Author):

In this paper, Jespersen et al. investigate the phylogenies of bacteria from a large number of metagenomes. More than 3,000 metagenome assembled genomes are assembled and taxonomically annotated. Based on 41 MAGs found in multiple samples, the authors investigate their variability as a function of their geographical location. They also investigate the functional origin of this variability. One of the main claims of this study is that global differences are caused by regional environmental selection. This conclusion is however highly over-optimistic. Based on the >3000 assembled MAGs as few as 41 were found in ten or more samples. Of these were 33 found in two geographical regions. Furthermore, of these were only 12 MAGs found to be significantly correlated with regional origin. Examining these 12 MAGs (Supplementary Figure 4), it is clear that several of them contain outliers (especially C1, C2, C14, C16, C17, C24). These outliers can very well come from other species and their inclusion into the MAG is highly dependent on the cut-off (the authors use a relaxed cut-off of a 95% ANI here). There are thus only 6 MAGs for which the geographical distribution can be assessed with any confidence (C3, C4, C5, C8, C12, C13). This is a fundamental flaw of the study – how can any general conclusion of genetic diversity and selection pressures acting on the sewage bacterial communities based on six species? The authors need, thus, to

a) Show that the results are consistent with a more relaxed and restricted MAG clustering cut-off. This is to make sure that their effects are indeed true any spurious based on the imperfect clustering. Performing the analysis with a relaxed cut-off is especially important since it should generate more MAGs that are present in multiple samples.

b) Show that their significant correlation with geographical distribution is still present when clear outliers are removed. This can easily be done by pruning the existing trees. The removal of regions represented by a single MAG is not sufficient to make sure that outliers do not affect your test (lines 368-369).

Furthermore, the authors only correct the p-values from the PERMANOVA tests but not for the WMW-test comparing the gene variance. Indeed, it is clear from SI Data 2 that several of those tests would become insignificant if proper correction for multiple testing was applied.

The final paragraph of the results (lines 228-233) needs to be rephrased. The conclusions reached here are both unclear and, as far as I can understand, not correct. The authors do not know which selection pressures act on the different gene groups and use a method with clearly limited statistical power and should therefore refrain to reach such general conclusions without further evidence.

Based on the issues raised above, the authors should revisit their abstract and conclusion. In several places, the conclusions of this study are highly overstated. For example, the abstract claims that "Applying these methods, we recovered 3,353 near complete (NC) metagenome assembled genomes (MAGs) encompassing 1,439 different MAG species and found that within species genomic variation was often coherent with regional separation." – a statement that is only true for a very small proportion of those MAGs. Similarly, the conclusion states that "In conclusion, this study shows a clear geographical phylogenetic clustering of bacterial species from sewage and suggests that this is caused by selection pressure." which simply can not be supported by the results presented in this study.

Additional comments

Lines 94-98, lines 134-136: It is unclear how these numbers affect the complexity of the binning algorithm. Binning communities with very similar strains are not necessarily more complicated than a diverse community. Please elaborate and provide some performance measures to show that the

binning is indeed less accurate or rephrase this sentence.

Lines 103-110: This part of the paragraph is confusing and needs to be clarified. The statement above is above 16S rRNA markers but the discussion seems to deal with species identification based on mapping to existing reference genes compare to genomes assembled de novo. Considering the many other factors influencing these numbers (e.g. difference in genome size, presence of high volumes of extracellular, strain-coverage in the reference genome database) the authors should tone down their statement ('due to the limitations of each of the different methods', line 103-104) or provide a data/references that support their speculations.

Lines 118-121: The KS test assesses the differences between samples – not two groups of samples. Please clarify how you did this test to also incorporate the large within-group variability.

Lines 128-131: The Shannon diversity depends on the sequencing depth. How was this corrected? I can't find any such details in the Method.

Lines 131-133: I do not understand how this sentence should be interpreted. How do you estimate "70% of the diversity of the human samples"? Do you by diversity mean Shannon diversity index? Also, what should the mapping of 41% of the sewage reads be compared to? Please clarify.

Lines 165-167: Please provide FDR significance cut-off. Based on the p-values in the SI Data 1, the cut-off needs to be at least $FDR < 0.1$ (since the smallest p-value is 0.003 and you made 33 tests). Provide also some argument for using such a high FDR.

Lines 204-210: These sentences imply a lack of evidence, not the evidence of the absence of an effect. Your approach may lack the power to see the effect and given the relatively low p-values throughout this study, it is certainly possible. Please rephrase.

Lines 342-343: Please provide the cut-offs used in the functional annotation.

Lines 369-370: "Benjamini & Hochberg" -> "Benjamini & Hochberg algorithm".

Lines 370-372: Was this done prior to the FDR estimation? If so, they need to be included. The FDR estimates the proportion of false discoveries based on the total number of performed tests. Performing tests and removing them before the FDR estimation will result in an incorrect (too optimistic) FDR.

Figure 2: Is the figure in panel d incorrect? The legend says "Only one gene involved in oxidative phosphorylation is found in all MAG species from sewage, whereas up to five genes from this pathway are found in the human gut MAG species." but the panel d tells a different story.

Supplementary Figure 1: There are too many similar colors in the figure in panel b) which makes it impossible to map the proportions to the correct phyla. Please reduce the number of phyla and make sure that the figure can be properly interpreted.

SI Data 3: The annotation at the species level for several MAGs does not include species information. Were these included in the statistics on lines 87-89. Burkholderiales is incorrectly annotated as Gammaproteobacteria (should be Betaproteobacteria).

Reviewer #2 (Remarks to the Author):

I reviewed this article previously for Nature Micro. I think the article is better suited to Communications Biology and will be of interest to the readership. I am satisfied that the authors have addressed most of my comments from the previous round of review. Although I do think the text could be further streamlined to assist in readability. Further comments follow.

Given much is made of the significance of this work (in gaining an insight into dispersal mechanisms for sewage related microbes) in the study of the spread of ARGs, I am curious as to why the authors didn't look into this here at all. I think many readers will find this strange.

Thank you for the comment and trying to increase readability of the manuscript. In our opinion this manuscript contains basic research focused on the within bacterial species phylogeny of sewage bacteria, their global spread, and whether potential selective mechanisms are impacting their global diversity. The spread of ARGs in sewage is investigated by us in other publications (Hendriksen et al. 03 08, 2019; Munk et al. 2022). In previous studies the global diversity of bacteria has mainly been studied at the genus level. We see this paper as a foundation that can be used to discuss/understand results of investigating also ARGs and their relation to different bacterial species and clonal strains. We mention the ARGs in the paper, as it is a reason why it is important to do these basic investigations on global diversity within bacterial species.

Line 83 - Define NC and MAG.

“NC MAGs” were changed to “near complete (NC) metagenome assembled genomes (MAGs)”

Line 87 - Which version of GTDB was applied? The later versions give species annotations. If these versions were used, is it the case that only 20% received annotation to this level?

The version of GTDB-Tk used is v0.3.2. It did give species annotations when possible. Only 699 MAGs corresponding to 20.8% of the identified MAGs were annotated at this level.

The version of GTDB-Tk is mentioned in the methods section line 296: “We assessed the taxonomy of all NC MAGs with GTDB-Tk classify_wf (v0.3.2).”

Lines 103-110 - I found the logic here to be vague and hard to follow. It is also likely due to the resolution differences between 16S and metagenomics and the cut-off values for similarity selected. Difficult to compare a figure like this.

The paragraph (now line 105-108) has been reduced and rewritten from:

“The difference in these results is most likely due to the limitations of each of the different methods for bacterial identification. Even though mapping of reads is restricted to the contents of reference databases, this method has the advantage of needing less coverage for a read to map, compared to the high read coverage that is necessary for a successful de novo assembly, which is a prerequisite to genome binning. Due to this high coverage need, genome binning has limitations in detecting low-abundance species. Thus, there are advantages and disadvantages for all methods and genome binning can be used to detect prevalent, including novel, bacterial genomes.”

To:

“The difference in these results could be due to the limitations of each of the different methods for bacterial identification. There are advantages and disadvantages for different methods for bacterial identification and genome binning can be used to detect prevalent, including novel, bacterial genomes.”

Lines 112-144 & 146-181 - Long paragraphs and in places hard to follow. Consider breaking up and streamlining the text.

We have tried to rewrite larger parts of the entire results-section to shorten sentences and clarify. This has reduced the results section from 2212 words to 1796 words. The changes are too extensive to be described separately, instead look in the results section of the attached manuscript. The major changes start from line 110.

Lines 199-200 - This sentence is not well integrated.

As part of the major revision of the results section, this sentence has been removed and the entire section on organelle gene variation has been rewritten.

Genus names should start with a capital letter and be italicized. i.e. *Salmonella*.

Thank you. This has been corrected in the text.

Line 235 - I don't think "comprehensive" is the right word here. Although a lot of work was done in this study, to me comprehensive would have involved more than just inference from metagenomic analyses. I would remove/rephrase.

The word “comprehensive” has been deleted from this sentence.

Lines 261-262 - "caused by selection pressure" is a bit vague. I think this should be rephrased.

Based on this comment and comments from the other reviewer this sentence has been modified. To clarify the expression has been elaborated, and the sentence (lines 251-253) has been changed from:

“In conclusion, this study shows a clear geographical phylogenetic clustering of bacterial species from sewage and suggests that this is caused by selection pressure.”

to:

“In conclusion, this study shows a clear geographical phylogenetic clustering of 12 bacterial species from sewage and suggests that this could be caused by global differences in the selection pressure in waste water and the corresponding adaptation of sewage bacteria.”

Lines 581-582 - Is this not the other way around? i.e. Only one gene in the human...

Thank you for this note. It is, as expected, the human gut species that lack genes involved in oxidative phosphorylation. We have corrected the figure legend (lines 593-595), so it now says:

“Only one gene involved in oxidative phosphorylation is found in all human gut MAG species, whereas up to five genes from this pathway is found in the MAG species from sewage.”

Line 596 - I found the interchangeable use of C5 and Brachymonas here to be confusing. Also, in the main text you have spelled it "Brahymonas".

To avoid confusion on this, we have corrected the figure legend to include “*Brachymonas (C5)*” every time this species is mentioned. Also, the spelling in the main text has been corrected, thank you for noticing.

Reviewer #3 (Remarks to the Author):

In this paper, Jespersen et al. investigate the phylogenies of bacteria from a large number of metagenomes. More than 3,000 metagenome assembled genomes are assembled and taxonomically annotated. Based on 41 MAGs found in multiple samples, the authors investigate their variability as a function of their geographical location. They also investigate the functional origin of this variability. One of the main claims of this study is that global differences are caused by regional environmental selection. This conclusion is however highly over-optimistic. Based on the >3000 assembled MAGs as few as 41 were found in ten or more samples. Of these were 33 found in two geographical regions. Furthermore, of these were only 12 MAGs found to be significantly correlated with regional origin. Examining these 12 MAGs (Supplementary Figure 4), it is clear that several of them contain outliers (especially C1, C2, C14, C16, C17, C24). These outliers can very well come from other species and their inclusion into the MAG is highly dependent on the cut-off (the authors use a relaxed cut-off of a 95% ANI here). There are thus only 6 MAGs for which the geographical distribution can be assessed with any confidence (C3, C4, C5, C8, C12, C13). This is a fundamental flaw of the study – how can any general conclusion of genetic diversity and selection pressures acting on the sewage bacterial communities based on six species? The authors need, thus, to

a) Show that the results are consistent with a more relaxed and restricted MAG clustering cut-off. This is to make sure that their effects are indeed true any spurious based on the imperfect clustering. Performing the analysis with a relaxed cut-off is especially important since it should generate more MAGs that are present in multiple samples.

First, we apologize that the number of species analyzed were in some places unclear. We have tried to change the text to make it clear that the geographical differences were only found for a few of the identified MAG species. These changes are described under some of the comments below.

We know that a species boundary of 95% ANI can be debated and understand the concerns of this reviewer. We therefore repeated the analysis with a more relaxed threshold of 90% and a more restricted of 97.5%. By doing this we identified few variations to the already identified MAG species, and only a single new MAG species.

The results have been included in the supplementary material of the manuscript as follows:

Main text (lines 137-141):

“Altering the 95% species ANI threshold to 90% and 97.5% only resulted in a few variations to the originally identified MAG species cluster sizes and only a single new MAG species, found in ten or more samples, was identified (Supplementary Figure 2). This is consistent with other studies suggesting a real species boundary around 95% ANI^{29,30}, although, these results are debated^{31,32}”.

New supplementary figure 2:

Supplementary Figure 2. Testing different species ANI thresholds. We tested the species threshold of 95% ANI, by running the analysis with different species thresholds. By increasing the threshold (97.5%) we identified one cluster subset (C2sub) which could be tested, and four subsets that did not meet our requirements for PERMANOVA testing. This subset did not cluster significantly according to geography (Supplementary Data 2). When decreasing the species ANI threshold, we identified five MAG species that differed from the initially identified species. Of these five, three comprised an original MAG species plus one or more additional MAGs (C2+, C5+, and C30+), one was a merge of two original MAG species (C27C28) and the last one was a new MAG species, not seen in the original analysis (C42). All the trees contained a group of samples (corresponding to an original MAG species, in the cases of C2+, C5+, C30+, and C27C28) and one or more outliers. Geographical clustering was significant in one of these trees (C2+), in which most of the outliers were from the same region. Taken together, these results suggest that there are not many new MAG species to be found, even though the ANI thresholds were decreased. In most of the cases, where new MAG species were identified, the novelty was based on outliers, potentially from the collapse of different bacterial species, which is also likely the case for C27C28.

PERMANOVA results added to Supplementary Data 2:

Different ANI species threshold						
	MAGs in tree	p value	p adjust	R2 geography	F	df
C2sub	24	0.017	0.051	0.317	2.206	4
C2+	36	0.001	0.006	0.527	6.675	5
C5+	38	0.984	0.984	0.049	0.423	4
C30+	14	0.906	0.984	0.148	0.577	3
C27C28	14	0.237	0.356	0.269	1.229	3
C42	8	0.122	0.244	0.431	4.544	1

b) Show that their significant correlation with geographical distribution is still present when clear outliers are removed. This can easily be done by pruning the existing trees. The removal of regions represented by a single MAG is not sufficient to make sure that outliers do not affect your test (lines 368-369).

The reviewer has written that: “Examining these 12 MAGs (Supplementary Figure 4), it is clear that several of them contain outliers (especially C1, C2, C14, C16, C17, C24).”. We wanted to accommodate the reviewer's suggestion, however, some of the listed MAG species were not included in the re-analysis due to the following: C16 is not one of the MAG species with significant geographical clustering and in the tree of C17 we cannot identify any outliers (See figure below). These trees have therefore not been pruned and re-tested. For the remaining four trees (C1, C2, C14, and C24) we did remove the outliers (highlighted in red in the figure below). Additionally, we removed an outlier from the tree of C4 and included this in the reanalysis as well.

In this figure we have included all the trees with significant geographical clustering and highlighted with red circles, the samples that was removed by pruning the trees:

C14 could not be reanalyzed after pruning of the tree, as there are no longer “two regions with at least two samples” if we remove one of the “ends” from the tree.

The results have been included in the supplementary as follows:

Main text, line 167: “These results were consistent when pruning trees from outliers (Supplementary Figure 6).”

Supplementary Figure 6. Outlier removal. In five trees (C1, C2, C4, C14, and C24) with significant geographical clustering outliers were identified. In one case (C14), the tree consisted of two groups of MAGs and could not be tested, if one of the groups were removed. The four remaining trees were pruned from outliers and re-tested with PERMANOVA. In three cases (C1prun, C2prun, and C4prun) the geographical clustering was still significant after pruning of the trees (Supplementary Data 2). In the last case (C24prun) the tree was reduced to comprise five MAGs, because the initial tree consisted of two groups of MAGs. Taken together, 10 out of the 12 significant MAG species trees do not have outliers or are still significant after pruning, the last two trees are potentially comprising distinct subgroups within the species.

PERMANOVA results added to Supplementary Data 2:

Pruning of trees						
	MAGs in tree	p value	p adjust	R2 geography	F	df
C1prun	43	0.001	0.002	0.643	17.111	4
C2prun	23	0.002	0.003	0.492	4.363	4
C4prun	36	0.001	0.002	0.306	4.698	3
C24prun	4	0.333	0.333	0.377	1.211	1

Furthermore, the authors only correct the p-values from the PERMANOVA tests but not for the WMW-test comparing the gene variance. Indeed, it is clear from SI Data 2 that several of those tests would become insignificant if proper correction for multiple testing was applied.

We apologize that the multiple test correction was not clear from the materials and methods section. We did correct all p-values (also from Wilcoxon and Kruskal Wallis tests) with BH correction. This was mistakenly not included in the methods and only in the figure legends. A sentence has been added to the methods section to describe this (line 381-382):

“P-values from the Wilcoxon- and Kruskal-Wallis tests were adjusted using the BH algorithm⁶⁰, adjusting for the number of gene group tests.”

Additionally, a sentence on the number of test that was adjusted by, were added to lines 371-374:

”To adjust for multiple testing, we corrected the p-values from these tests using Benjamini & Hochberg (BH) algorithm⁶⁰, **adjusting for the number of ASTRAL species trees tested with PERMANOVA.**”

Furthermore, SData 2 has been updated to include both initial p-values and adjusted p-values for all tests.

The final paragraph of the results (lines 228-233) needs to be rephrased. The conclusions reached here are both unclear and, as far as I can understand, not correct. The authors do not know which selection pressures act on the different gene groups and use a method with clearly limited statistical power and should therefore refrain to reach such general conclusions without further evidence.

Again, we apologize for the left-out description on the multiple test corrections. We do believe that the investigations on R2 values in different gene groups, supported by the dN/dS values in the same gene groups support a suggestion on regional selection for the species with significant differences in these measurables. To make it clear, that this suggestion is only applicable for a selection of species we have added some words to these lines (line 219-224):

Previous text:

“Collectively, our results show similar levels of geographical clustering in the membrane genes and metabolic genes, whereas the genes associated with the organelles displayed significantly less geographical clustering. We expect that the organelle genes would have followed a similar evolutionary trajectory as the metabolic and surface genes if the clustering was a result of dispersal limitations. Our

results thus suggest that the geographical clustering is primarily due to regional selection and not dispersal limitations.”

New text:

*“Collectively, our results show similar levels of geographical clustering in the membrane genes and metabolic genes **for a selection of species**, whereas the genes associated with the organelles displayed significantly less geographical clustering. We expect that the organelle genes would have followed a similar evolutionary trajectory as the metabolic and surface genes if the clustering was a result of dispersal limitations. Our results thus suggest that **for some species (36% of the investigated)** the geographical clustering is primarily due to regional selection, rather than dispersal limitations.”*

Based on the issues raised above, the authors should revisit their abstract and conclusion. In several places, the conclusions of this study are highly overstated. For example, the abstract claims that *“Applying these methods, we recovered 3,353 near complete (NC) metagenome assembled genomes (MAGs) encompassing 1,439 different MAG species and found that within species genomic variation was often coherent with regional separation.”* – a statement that is only true for a very small proportion of those MAGs. Similarly, the conclusion states that *“In conclusion, this study shows a clear geographical phylogenetic clustering of bacterial species from sewage and suggests that this is caused by selection pressure.”* which simply can not be supported by the results presented in this study.

Thank you for pointing this out. We understand that the abstract can be interpreted as the findings showing regional separation for most of the identified MAG species. We have changed the wording in the abstract and conclusion to clarify that this is only found for some species.

Changes in line 24-26 in the abstract from:

“Applying these methods, we recovered 3,353 near complete (NC) metagenome assembled genomes (MAGs) encompassing 1,439 different MAG species and found that within species genomic variation was often coherent with regional separation.”

To:

*“Applying these methods, we recovered 3,353 near complete (NC) metagenome assembled genomes (MAGs) encompassing 1,439 different MAG species and found that within-species genomic variation was in **36% of the investigated species (12/33)** coherent with regional separation.”*

And in lines 30-32:

“From the combination of the large and globally distributed dataset with the in-depth analysis methods, we present the most comprehensive investigation of global within-species phylogeny from metagenomics data to date”

has been corrected to:

“From the combination of a large and globally distributed dataset and an in-depth analysis, we present a wide investigation of global within-species phylogeny of microbial sewage species.”

Also, in lines 72-76: in the introduction some words were added to clarify that the geographical clustering were only found in some of the MAG species:

*“From our analysis, we identified 3,353 near complete (NC) MAGs from 1,439 different MAG species and found that variation within **some species (12 species out of 33 investigated)** correlated with*

geographical separation. Furthermore, we found that, for a selection of species, genes associated with organelles displayed on average 10% less geographical variation compared to other groups of genes, suggesting that the geographical clustering is primarily due to environmental selection.”

Additionally, lines 252-254 of the discussion has been changed from:

“In conclusion, this study shows a clear geographical phylogenetic clustering of bacterial species from sewage and suggests that this is caused by selection pressure.”

to:

*“In conclusion, this study shows a clear geographical phylogenetic clustering of **12** bacterial species from sewage and suggests that this **could be** caused by global differences in the selection pressure in wastewater and the corresponding adaptation of sewage bacteria.”*

Additional comments

Lines 94-98, lines 134-136: It is unclear how these numbers affect the complexity of the binning algorithm. Binning communities with very similar strains are not necessarily more complicated than a diverse community. Please elaborate and provide some performance measures to show that the binning is indeed less accurate or rephrase this sentence.

From the Critical Assessment of Metagenome Interpretation 2 (CAMI2) it was found that low assembly quality and strain diversity introduced binning challenges and reduced performance. Likewise, we found that highly similar strains (ANI > 99%) were at risk of being mixed into one MAG, when testing for the effect of strain mixing in this manuscript (Supplementary Figure 3 and Supplementary Table 1). We do believe that this is reflected in our binning results and results in a lower number of NC MAGS, that are, however, assigned to a higher number of different MAG species. Additionally, we saw that the percentage of reads mapping to the MAG species identified within the samples were higher for the human samples than for the sewage samples. Collectively, we think that these results suggest that the binning of the sewage samples has been more challenging. We have included the CAMI2 paper as a reference and changed the sentence (Line 95-100) from:

“The 3,353 NC MAGs we identified were less than the number of NC MAGS (5,036) found from binning of 1,000 human faecal samples, however, these gut MAGs represented a lower number of different MAG species (645), suggesting that the binning of sewage metagenomes is more complicated than binning of human gut samples.”

To:

“The 3,353 NC MAGs we identified were less than the number of NC MAGS (5,036) found from binning of 1,000 human faecal samples, however, these gut MAGs represented a lower number of different MAG species (645). Lower assembly quality and strain diversity has been suggested to reduce binning performance^s, which could be why the binning of these sewage metagenomes is more complicated than binning of the human gut samples.”

The previous lines 134-136 were part of the major rewriting of the results section and are now part of lines 130-132:

“Together our results suggest a major shift in taxonomy, enrichment for aerophilic organisms and a

lower genome retrieval efficiency, due to higher alpha diversity and multi-host strain heterogeneity, in the sewage compared to human guts.”

Lines 103-110: This part of the paragraph is confusing and needs to be clarified. The statement above is above 16S rRNA markers but the discussion seems to deal with species identification based on mapping to existing reference genes compare to genomes assembled de novo. Considering the many other factors influencing these numbers (e.g. difference in genome size, presence of high volumes of extracellular, strain-coverage in the reference genome database) the authors should tone down their statement (‘due to the limitations of each of the different methods’, line 103-104) or provide a data/references that support their speculations.

The paragraph (lines 105-108) has been rewritten and partly deleted from:

“The difference in these results is most likely due to the limitations of each of the different methods for bacterial identification. Even though mapping of reads is restricted to the contents of reference databases, this method has the advantage of needing less coverage for a read to map, compared to the high read coverage that is necessary for a successful de novo assembly, which is a prerequisite to genome binning. Due to this high coverage need, genome binning has limitations in detecting low-abundance species. Thus, there are advantages and disadvantages for all methods and genome binning can be used to detect prevalent, including novel, bacterial genomes.”

To:

*“The difference in these results **could be** due to the limitations of each of the different methods for bacterial identification. There are advantages and disadvantages for different methods for bacterial identification and genome binning can be used to detect prevalent, including novel, bacterial genomes.”*

Lines 118-121: The KS test assesses the differences between samples – not two groups of samples. Please clarify how you did this test to also incorporate the large within-group variability.

Thank you for this comment. We did not incorporate the within-group variability in the initial test. Now we have changed the test to test each region of sewage bacteria separately against the set of human bacteria. The results of this new test can be found in the last sheet in S Data 2. We have also updated the text with a reference to these test statistics (line 117).

Lines 128-131: The Shannon diversity depends on the sequencing depth. How was this corrected? I can’t find any such details in the Method.

We apologise that this information was not found in the methods section. The abundances found with CoverM were obtained as Transcripts Per Kilobase Million (TPM) and thus corrected for the sequencing depth of each sample.

The lines 302-305 in the methods section have been corrected from:

“The abundances of MAG species were obtained using CoverM (v 0.6.1)³⁷, by mapping to the best representative MAG species genomes and filtering based on the expected/covered ratio of a genome (≥ 0.5).”

To:

*“The abundances of MAG species were obtained as **Transcripts Per Kilobase Million (TPM)** using CoverM (v 0.6.1)³⁸, by mapping to the best representative MAG species genomes and filtering based on the expected/covered ratio of a genome (≥ 0.5).”*

Lines 131-133: I do not understand how this sentence should be interpreted. How do you estimate “70% of the diversity of the human samples”? Do you by diversity mean Shannon diversity index? Also, what should the mapping of 41% of the sewage reads be compared to? Please clarify.

We apologize that this sentence was confusing. We simply meant that 70% of the reads mapped to the human MAG species. We have now changed diversity to reads in the sentence (lines 128-129), from:

“Furthermore, when mapping reads to all the MAG species identified in either human gut or sewage samples, the MAG species captured around 70% of the diversity of the human samples, whereas only 41% of the sewage reads were mapped.”

To:

“Furthermore, around 70% of the human gut microbiome reads, whereas just 41% of the sewage reads, were assigned to the MAG collection.”

Lines 165-167: Please provide FDR significance cut-off. Based on the p-values in the SI Data 1, the cut-off needs to be at least $FDR < 0.1$ (since the smallest p-value is 0.003 and you made 33 tests). Provide also some argument for using such a high FDR.

We assume that you get this FDR based on p-value adjustment (33×0.003). As previously written, the p-values in SI data 2 (SI data 1 is the taxonomy table), have been corrected using BH. The cutoff was $P < 0.05$ and this has been added to lines 163-164. Additionally, both uncorrected and BH adjusted P values are now included in SI Data 2.

“Of the 33 tested trees, 12 clustered significantly ($P < 0.05$, BH corrected) according to genome sample region (Figure 3b).”

Lines 204-210: These sentences imply a lack of evidence, not the evidence of the absence of an effect. Your approach may lack the power to see the effect and given the relatively low p-values throughout this study, it is certainly possible. Please rephrase.

Again, we apologize for the p-value confusion. During the rewriting of the results section, the sentence was changed from:

“Thus, differences in R2 values found between gene groups was not a result of a reduced variation of the organelle genes. In addition to this, we tested for the effect of different gene lengths, as well as the

number of MAGs and the distribution of different regions in a tree. For one MAG species (C26) we found that the R² values correlated with the number of MAGs in the tree and regional entropy, therefore, this cluster was excluded from further analyses. For the remaining 32 MAG species, none of these factors correlated with the R² values, meaning that they were not relevant to include in subsequent statistical testing.”

To:

“As we found that organelle genes varied less than other genes in 94% of species, we wanted to ensure that our test was not driven by differences in gene conservation levels. This was not the case, as gene variation was unrelated to the R² values (Supplementary Figure 6). Additionally, gene lengths, regional distributions within trees, and the number of MAGs per species were also unrelated to geographical clustering, with the exception of one species (C26), which we therefore excluded from further analysis.”

Lines 342-343: Please provide the cut-offs used in the functional annotation.

We used the output from Interproscan as function annotations directly, without applying any cutoffs. We believe this is the correct way to use this output according to the interproscan documentation (https://interproscan-docs.readthedocs.io/_/downloads/en/latest/pdf/).

This is the section of the Interproscan documentation on match cutoff:

“The e-values are specific to each individual InterPro member database and therefore cannot be compared directly, or a single threshold applied to them all. This is because some member databases use the e-values for post-processing (e.g. SMART, Panther), others just output it as part of their results but actually use other measures for filtering of results (e.g. Pfam and the Hmmer GA cut-off). Therefore as far as InterProScan is concerned, if a match is in the output then it is a match!”

Lines 369-370: “Benjamini & Hochberg” -> “Benjamini & Hochberg algorithm”.

Corrected.

Lines 370-372: Was this done prior to the FDR estimation? If so, they need to be included. The FDR estimates the proportion of false discoveries based on the total number of performed tests. Performing tests and removing them before the FDR estimation will result in an incorrect (too optimistic) FDR.

To be clear, none of the species tree R² values were negative. The negative R² values were only found in a small number (<0.5%) of the gene tree PERMANOVA tests. From the gene tree PERMANOVA tests we grouped the geographical R² values based on gene product cellular location/function (membrane, organelle, metabolic) and tested for differences between these groups (initially, Kruskal-Wallis test followed by Wilcoxon, both BH corrected). Thus, the removal of the R² values only changes the underlying data that were grouped and tested, not the number of tests.

However, this comment made us think about the exclusion of these values, and we decided to include them in the analysis. Therefore, negative R² values are now included in the gene group analysis and

figures 3c, 3d, supplementary figure 9, and supplementary data 2 have been changed accordingly. The inclusion of the negative R2 values did not change which species had significant differences between gene groups. The conclusions of these tests are thus the same as previously.

Figure 2: Is the figure in panel d incorrect? The legend says “Only one gene involved in oxidative phosphorylation is found in all MAG species from sewage, whereas up to five genes from this pathway are found in the human gut MAG species.” but the panel d tells a different story.

The figure is correct, but the figure legend was faulty. It is, as expected, the human gut species that lack genes involved in oxidative phosphorylation. We have corrected the figure legend, so it now says:

“Only one gene involved in oxidative phosphorylation is found in all human gut MAG species, whereas up to five genes from this pathway is found in the MAG species from sewage.”

Supplementary Figure 1: There are too many similar colors in the figure in panel b) which makes it impossible to map the proportions to the correct phyla. Please reduce the number of phyla and make sure that the figure can be properly interpreted.

This supplementary figure included all the phyla found in the data. A reduced version of the figure can be found in figure 2a in the manuscript. The supplementary figure 1b has been removed from the manuscript.

SI Data 3: The annotation at the species level for several MAGs does not include species information. Were these included in the statistics on lines 87-89. Burkholderiales is incorrectly annotated as Gammaproteobacteria (should be Betaproteobacteria).

Thank you for noticing this. By fault, some species names were only partly transferred from the GTDB-Tk output to the supplementary. This has now been corrected and the species names should be complete. The proportions of unannotated species (and genus etc.) have been double checked and are correct. Additionally, Gammaproteobacteria has been corrected to Betaproteobacteria for all Burkholderiales

References

- Hendriksen, Rene S., Patrick Munk, Patrick Njage, Bram van Bunnik, Luke McNally, Oksana Lukjancenko, Timo Röder, et al. 03 08, 2019. "Global Monitoring of Antimicrobial Resistance Based on Metagenomics Analyses of Urban Sewage." *Nature Communications* 10 (1): 1124.
- Munk, Patrick, Christian Brinch, Frederik Duus Møller, Thomas N. Petersen, Rene S. Hendriksen, Anne Mette Seyfarth, Jette S. Kjeldgaard, et al. 2022. "Genomic Analysis of Sewage from 101 Countries Reveals Global Landscape of Antimicrobial Resistance." *Nature Communications* 13 (1): 1–16.

Reviewers' comments:

Reviewer #4 (Remarks to the Author):

I have previously reviewed this paper. The author has made a significant effort to revise the manuscript and tone down many of the over-optimistic interpretations. Most of my raised concerns have therefore been addressed. There are, however, two issues that need to be addressed before publication.

The correction for sequencing depth for estimation of the Shannon diversity. Normalization based on the total counts does not remove the difference in detection limit between samples with different sequence depths. The authors need to rarefy their data so that samples have a similar sequence depth to make the diversity index comparable.

I am happy that the authors were able to show that their results were robust with respect to the ANI cut-off parameter. The remark made by the authors in the supplement, i.e. "This is consistent with other studies suggesting a real species boundary around 95% ANI, although, these results are debated.", is, however, misleading. The issue with setting the ANI value globally is that it is known to vary substantially between different parts of the taxonomic tree – there is thus no "real species boundary" that can be defined by a single ANI cut-off. Please rephrase to make this clear.

Reviewers' comments:

Reviewer #4 (Remarks to the Author):

I have previously reviewed this paper. The author has made a significant effort to revise the manuscript and tone down many of the over-optimistic interpretations. Most of my raised concerns have therefore been addressed. There are, however, two issues that need to be addressed before publication.

The correction for sequencing depth for estimation of the Shannon diversity. Normalization based on the total counts does not remove the difference in detection limit between samples with different sequence depths. The authors need to rarefy their data so that samples have a similar sequence depth to make the diversity index comparable.

Thank you for this comment. We have now rarefied our data using the `rrarefy` function from the `vegan` package in R. This did not change the overall Shannon diversity distributions for any of the geographic groups or the human gut samples, probably because we did previously also filter the abundances based on the expected/observed coverage ratio (≥ 0.5).

We have corrected the mean Shannon diversity in line 125-130 of the results section:

“Additionally, these results suggest that species associated with humans make up a minor proportion of a taxonomically more diverse sewage bacteriome with more potential sources (mean Shannon diversity of >3.75 and 2.4 in sewage and human gut, respectively, Figure 2d).”

We added the rarefaction step to line 306-313 of the methods section:

“The read count abundances of MAG species were obtained using `CoverM` (v 0.6.1)41, by mapping to the best representative MAG species genomes. The `rrarefy` function from the `Vegan` R package was used to rarefy the counts, afterwards the abundance matrix was filtered based on the expected/covered ratio of a genome (≥ 0.5) and Transcripts Per Kilobase Million (TPM) were calculated from the read counts.”

And modified figure 2 with the new Shannon diversity plot:

I am happy that the authors were able to show that their results were robust with respect to the ANI cut-off parameter. The remark made by the authors in the supplement, i.e. “This is consistent with other studies suggesting a real species boundary around 95% ANI, although, these results are debated.”, is, however, misleading. The issue with setting the ANI value globally is that it is known to vary substantially between different parts of the taxonomic tree – there is thus no “real species boundary” that can be defined by a single ANI cut-off. Please rephrase to make this clear.

This is a good point and we have tried to include it in lines 140-141:

“This is consistent with other studies suggesting the use of 95% ANI as a species threshold^{29,30}, however, ANI boundaries could vary between species and these results are debated^{31,32}”

REVIEWERS' COMMENTS:

Reviewer #4 (Remarks to the Author):

The authors have addressed all my comments. I only have one minor comment before the publication.

Please add the sequencing used for the rarefaction of the data.

REVIEWERS' COMMENTS:

Reviewer #4 (Remarks to the Author):

The authors have addressed all my comments. I only have one minor comment before the publication.

Please add the sequencing used for the rarefaction of the data.

We believe, the suggestion from the reviewer is to add the sequencing *depth* of the rarefaction and have updated lines 305-308 to include this:

*“The rrarefy function from the Vegan R package was used to rarefy the counts **to the minimum number of counts (431.564) found in a sample**, afterwards the abundance matrix was filtered based on the expected/covered ratio of a genome (≥ 0.5) and Transcripts Per Kilobase Million (TPM) were calculated from the read counts.”*